# Strictinin, a Major Ingredient in Yunnan Kucha Tea Possessing Inhibitory Activity on the Infection of Mouse Hepatitis Virus to Mouse L Cells

**DOI:** 10.3390/molecules28031080

**Published:** 2023-01-21

**Authors:** Erh-Chuang Tu, Wei-Li Hsu, Jason T. C. Tzen

**Affiliations:** 1Graduate Institute of Biotechnology, National Chung-Hsing University, Taichung 402, Taiwan; 2Graduate Institute of Microbiology and Public Health, National Chung-Hsing University, Taichung 402, Taiwan

**Keywords:** Kucha Pu’er tea, mouse hepatitis virus, remdesivir, strictinin, theacrine

## Abstract

Theacrine and strictinin of Yunnan Kucha tea prepared from a mutant variety of wild Pu’er tea plants were two major ingredients responsible for the anti-influenza activity. As the COVID-19 outbreak is still lurking, developing safe and cost-effective therapeutics is an urgent need. This study aimed to evaluate the effects of these tea compounds on the infection of mouse hepatitis virus (MHV), a β-coronavirus serving as a surrogate for SARS-CoV. Treatment with strictinin (100 μM), but not theacrine, completely eliminated MHV infection, as indicated by a pronounced reduction in plaque formation, nucleocapsid protein expression, and progeny production of MHV. Subsequently, a time-of-drug addition protocol, including pre-, co-, or post-treatment, was exploited to further evaluate the possible mechanism of antiviral activity mediated by strictinin, and remdesivir, a potential drug for the treatment of SARS-CoV-2, was used as a positive control against MHV infection. The results showed that all three treatments of remdesivir (20 μM) completely blocked MHV infection. In contrast, no significant effect on MHV infection was observed when cells were pre-treated with strictinin (100 μM) prior to infection, while significant inhibition of MHV infection was observed when strictinin was introduced upon viral adsorption (co-treatment) and after viral entry (post-treatment). Of note, as compared with the co-treatment group, the inhibitory effect of strictinin was more striking in the post-treatment group. These results indicate that strictinin suppresses MHV infection by multiple mechanisms; it possibly interferes with viral entry and also critical step(s) of viral infection. Evidently, strictinin significantly inhibited MHV infection and might be a suitable ingredient for protection against coronavirus infection.

## 1. Introduction

Severe acute respiratory syndrome (SARS), a highly contagious respiratory disease caused by the SARS coronavirus (SARS-CoV), emerged in 2002, spread rapidly around the world, and ended in late 2003 with a mortality rate of 10–15% [1]. SARS-CoV is a zoonotic virus possibly propagated in animals, and thus the recurrence of SARS or the spread of similar diseases caused by mutated SARS-CoV seems inevitable. As predicted, the COVID-19 outbreak caused by a new type of β-coronavirus, named SARS-CoV-2, was first reported in Wuhan, China, in December 2019, and quickly spread to all parts of the world [2]. Several mutants of SARS-CoV-2 were subsequently detected as the prevailing strains in the past few years. Although its mortality rate (around 2%) was much lower than that of SARS, COVID-19 was found to be highly contagious, with many asymptomatic or mild patients in their first infection, which might turn out to be a great threat to human health in the following years [3].

Antiviral drugs, such as remdesivir, dexamethasone, lopinavir/ritonavir, and hydroxychloroquine/chloroquine, are generally recommended for the treatment and prevention of coronavirus diseases before specific medicines and vaccines are available; however, severe side effects were reported for patients treated with these antiviral drugs [4,5]. Herbal medicines have been used to heal many diseases, including those caused by viral infections, such as influenza and the common cold [6,7]. Therefore, natural compounds in herbs seem to be rich sources for screening potential drugs against coronaviruses. As expected, several herbal medicines as well as isolated compounds in herbs have been reported to possess inhibitory activities against SARS-CoV-2 in the past few years [8,9,10,11,12,13,14,15]. Although herbal medicines and their isolated compounds are not as effective as chemical antiviral drugs, they are empirically safe for the human body, particularly after long-term consumption.

Pu’er tea, prepared in a restricted area of Yunnan, China, has been consumed for thousands of years as an herbal tea with many beneficial effects for human health, such as antiviral, antimicrobial, anti-obesity, and anti-cancer activities [16,17]. Yunnan Kucha (bitter) tea prepared from a mutant variety of wild Pu’er tea plant is assumed to possess a superior anti-influenza activity according to the empirical utilization of indigenous aborigines in certain uncultivated mountain areas of Yunnan. Recently, theacrine and strictinin were identified as two major compounds responsible for the anti-influenza activity of Yunnan Kucha tea [18]. Theacrine is structurally similar to caffeine, but much more bitter than caffeine in taste, and has been demonstrated to possess several biological activities, including a sedative effect on sleep improvement, ergogenic effect on sports performance, anti-inflammation, anti-depression, and anti-breast cancer [19,20,21,22,23,24,25]. Strictinin is a hydrolyzable ellagitannin, generally detected as a minor constituent in most tea varieties but abundantly present in Pu’er tea, and has been demonstrated to possess antioxidant, antiviral, antimicrobial, anti-obesity, and anti-psoriatic activities [26,27,28,29,30].

It has not been elucidated if Yunnan Kucha may serve as an herbal tea for the supplementary treatment or prevention of coronavirus diseases, such as COVID-19. As it is known that SARS-CoV-2, the etiologic agent of COVID-19, is a risk group 3 (RG-3) pathogen, drug development has been hindered by the lack of BSL-3 facilities. Alternative strategies have been exploited to circumvent this limitation, including using pseudotyped virus [31] or murine hepatitis virus (MHV) as a surrogate model for the study of SARS-CoV entry or replication, respectively [32,33,34]. MHV belongs to the β-coronavirus, in the same subgenus of SARS-CoV-2, and it could be an excellent model to investigate the pathogenesis in the aspects of virulence and immune response to coronaviruses including SARS-CoV. Moreover, it provides some insight into the viral biology of SARS-CoV-2. Most importantly, unlike SARS-CoV-2, handling MHV infection only requires biosafety BSL-2 containment. Hence, an initial attempt was made to explore the inhibitory activity of theacrine and strictinin, two known anti-influenza compounds in Yunnan Kucha tea, on the infection of MHV, and the potential mechanism was further investigated by a time-of-drug addition test protocol in comparison with that of remdesivir.

## 2. Results

### 2.1. Cytotoxicity of Theacrine and Strictinin to Mouse L Cells

Cytotoxicity to mouse L cells was examined by the PrestoBlue staining method for theacrine and strictinin. The results showed that both compounds had no significant cytotoxicity to mouse L cells, at least up to a concentration of 100 µM (Figure 1). Therefore, the effects of these two compounds on MHV infection were evaluated with the highest concentration of 100 µM in the following experiments.

### 2.2. Effects of Theacrine and Strictinin on MHV Infection of Mouse L Cells

#### 2.2.1. Inhibitory Activity of Theacrine and Strictinin on Plaque Formation

The inhibitory activity of theacrine and strictinin on MHV infection was examined by a plaque reduction assay. Mouse L cells were infected with MHV in the presence of theacrine or strictinin at a concentration of 4, 20, or 100 μM in the course of infection. The results showed that no significant inhibition of plaque formation was observed for theacrine with a final concentration of up to 100 μM (Figure 2). In contrast, significant inhibition of plaque formation was observed for strictinin at a concentration of 20 μM and 100 μM; plaque formation was reduced to less than 25% of mock treatment, or completely inhibited in the presence of strictinin at a concentration of 20 μM, 100 μM, respectively.

#### 2.2.2. Effects of Theacrine and Strictinin on the Level of Nucleocapsid Protein

Mouse L cells were infected with MHV in the presence of theacrine or strictinin of 100 μM, and harvested at 12 and 24 h post-infection (hpi). The relative expression levels of nucleocapsid protein (NP) after MHV infection were detected by Western blot analysis. The results showed that no significant effect on viral NP expression was observed for the supplement of theacrine though a slight reduction (with no statistical significance) was observed for the cells harvested at 24 hpi (Figure 3). In contrast, a significant reduction in the level of NP was observed for cells harvested at 12 and 24 hpi in the presence of strictinin. Indeed, the level of NP was completely eliminated by the supplement of strictinin at a concentration of 100 μM in the culture medium.

#### 2.2.3. Effects of Theacrine and Strictinin on Viral Progeny Production

Subsequently, the relative viral yield of MHV in mouse L cells after infection was estimated by the plaque assay. As with the effect on NP protein, supplementation with theacrine did not have a significant impact on the viral progeny production, though a slight reduction (approximately 78.8% inhibition, with no statistical significance) was observed for the cells harvested at 24 hpi (Figure 4). In contrast, significant effects on viral progeny production were observed in infected cells treated with strictinin at a concentration of 100 μM. Indeed, at 12 hpi, the viral progeny production reduced by at least 1000 folds, and the inhibitory effect remained till 24 hpi.

### 2.3. Comparison of Effects of Remdesivir and Strictinin on MHV Infection of Mouse L Cells

According to the above observation, strictinin but not theacrine was subjected to further analysis of its inhibitory activity in different stages of MHV infection by the time-of-drug addition protocol [18]. The inhibitory activity of remdesivir (20 μM) on MHV infection was used as a positive control.

#### 2.3.1. Inhibitory Activity of Remdesivir and Strictinin on Plaque Formation

Mouse L cells were pre-treated, co-treated, or post-treated with remdesivir at 20 μM or strictinin at 100 μM during the MHV infection. The inhibitory activity of remdesivir and strictinin on MHV infection was initially examined by a plaque reduction assay. The results showed that remdesivir significantly inhibited MHV infection, as evidenced by the complete blockage of plaque formation in all three treatments (Figure 5). In contrast, no significant inhibition of plaque formation was observed in cells treated with strictinin 8 h prior to infection (pre-treatment group). However, significant inhibition (by approximately 40%) of plaque formation was observed for the co-treatment of strictinin upon viral attachment. Strikingly, plaque formation was completely abrogated in the post-treatment group, in which strictinin was supplemented to cells after viral entry (i.e., at 2 hpi).

#### 2.3.2. Effects of Remdesivir and Strictinin on the Level of Nucleocapsid Protein

Mouse L cells were infected with MHV in combination with the three treatments of remdesivir or strictinin and harvested at 12 and 24 hpi. The relative expression levels of viral NP were detected by Western blot analysis. Consistently, three different treatments of remdesivir led to a significant reduction in the accumulated level of NP in infected cells harvested at 12 and 24 hpi (Figure 6). In contrast, no significant effect on the NP level was observed for the pre-treatment of strictinin. However, a significant reduction in NP expression was observed for the co-treatment and post-treatment of strictinin. Relatively, the effect of post-treated strictinin on the reduction in NP was stronger than that of co-treated strictinin. Of note, at 12 hpi, expression of NP was completely abolished in the infected cell treated with strictinin after viral entry.

#### 2.3.3. Effects of Remdesivir and Strictinin on Viral Progeny Production

Ultimately, the overall viral yield of MHV in infected mouse L cells pre-treated, co-treated, or post-treated with remdesivir or strictinin was measured by the plaque assay. The results showed a significant reduction in viral progeny production in cells treated with remdesivir (Figure 7). In contrast, pre-treatment with strictinin had no significant effect on viral progeny production. However, a pronounced reduction in viral progeny production was observed in the other two strictinin groups (i.e., co-treatment and post-treatment). Consistent with effects on plaque reduction assay and NP expression, relatively, the effect of post-treated with strictinin on the reduction in viral progeny production was stronger than that of co-treated with strictinin. Treatment with strictinin upon viral attachment decreased the overall viral production to approximately 2.5% of that without treatment. Finally, the viral yield in the post-treatment of the strictinin group remained at only 0.1%, 0.2% of mock treatment, indicating a significant suppression of viral propagation.

## 3. Discussion

In a previous study, theacrine and strictinin, abundantly found in Yunnan Kucha tea, were demonstrated to possess anti-influenza activity [18]. In this study, the effect of these two compounds on the infection of MHV was monitored in three systems, including plaque formation ability as well as production of viral protein and viral progeny. Of note, strictinin but not theacrine was shown to effectively inhibit MHV infection of mouse L cells. As influenza viruses and MHV are different in many aspects, for instance, composed of different sense RNA segments, genome structure, and replication machineries, it is not too surprising that theacrine is able to inhibit influenza but not MHV. Presumably, theacrine and strictinin displayed their anti-influenza activities by way of distinct molecular mechanisms. Although theacrine was found unable to inhibit MHV infection directly, its other biological functions, such as the sedative effect on sleep improvement and enhancement of the immunological system in humans [19,21], should not be ignored as these functions might also be helpful for the recovery after viral infections.

MHV is a β-coronavirus, which is classified in the same subgenus as SARS-CoV and SARS-CoV-2. Owing to the sequence similarity and replication characteristics, MHV has been exploited as a safe surrogate virus for drug repurposing screens for combating COVID-19 [35]. Several repurposing drugs were found to consistently reduce or block the infection of both MHV and SARS-CoV-2. For instance, β-DN4-hydoxycytidine, an active form of ribonucleoside analog (molnupiravir), was effective for coronaviruses, including SARS-CoV, SARS-CoV-2, and MHV, by inhibiting replication at early stages of the viral replication cycle [36]. Similarly, Tan et al. reported that remdesivir, a well-known antiviral drug prescribed for COVID-19, could hamper MHV infection and the excessive cytokine production induced by MHV in murine RAW264.7 cells [35]. Hence, MHV was proposed as an alternative in vitro model to mimic macrophage infection in COVID-19 patients. Consistently, the study herein demonstrated significant anti-MHV activity of remdesivir in mouse L cells as measured by viral protein (Figure 7) and viral yield (Figure 7). Of note, treatment with strictinin upon viral attachment (co-treatment) and also after viral entry (post-treatment) remarkably abrogated the production of viral progenies, resulting in an up to 1000-fold reduction in the mock treatment group (Figure 7). No cytotoxic effect of strictinin, at least up to 100 μM, was detected, suggesting a safe and promising option for anti-coronavirus diseases.

Strictinin is thermally unstable and tends to be decomposed into ellagic acid and gallic acid at temperatures higher than 80 °C [27]. Indeed, strictinin, classified as an ellagitannin, is metabolized to ellagic acid in the human body [37]. Interestingly, ellagic acid, the converted product from strictinin, was found to possess higher inhibitory potency against human influenza virus A/Puerto Rico/8/34 than strictinin [27]. Recently, molecular modeling indicated that ellagic acid might be a potential compound having inhibitory activity against SARS-CoV [38], and the 3C-like protease of SARS-CoV was proposed to be the target of ellagic acid [39]. Moreover, a crude extract of Pu’er tea was found to inhibit the 3C-like protease of SARS-CoV-2, though the active ingredient was not identified [40]. Unfortunately, in the first screening test, we did not detect significant inhibitory activity of ellagic acid on the MHV infection in mouse L cells (data not shown). Whether ellagic acid is a suitable ingredient against various viruses remains to be clarified case by case in detail.

Pu’er tea has been empirically consumed as an effective herbal tea when people caught a cold regardless the symptoms were caused by the flu or common cold [18]. Possibly, strictinin, found exclusively abundant in Pu’er tea, is responsible for the broad effectiveness against various viral infections. Hydrolyzable tannin was also shown to bind to proteins non-specifically [41]. Ellagitannins, such as chebulagic acid and punicalagin, were found to execute broad-spectrum antiviral activities by interaction with glycoproteins on the surface of host cells to control viral infections [42]. Recently, chebulagic acid and punicalagin were shown to be novel allosteric inhibitors of SARS-CoV-2 3C-like protease [43]. Epigallocatechin gallate, the major catechin in tea, was also found to possess broad-spectrum antiviral activities through interacting with viral surface proteins to inhibit the attachment of many unrelated virions, such as herpes simplex virus type 1, hepatitis C virus, influenza A virus, vaccinia virus, adenovirus, reovirus, and vesicular stomatitis virus [44].

In this study, the possible anti-viral mechanism of strictinin was further interpreted by a time-of-drug addition assay [45]. As no significant effect of strictinin on MHV infection was observed when it was added to the medium of cell culture prior to infection but depleted through the time of viral infection (i.e., pre-treatment protocol), it implicates that strictinin is not sufficient to elevate the general anti-viral status. While co-treatment or post-treatment with strictinin elicited a pronounced inhibition or complete elimination of MHV infection, the presence of strictinin in these two treatments coincides with either the early stage of viral infection or after setting off the viral replication cycle in infected cells, respectively. The plaque formation in co-treatment remained at 60% of mock control (Figure 5), suggesting that viral entry was partially abolished by strictinin. Moreover, a stronger inhibitory effect of strictinin co-treatment was observed on viral NP protein and viral production. According to these results, it was very likely that strictinin could execute anti-MHV function extracellularly; it did not get into the mouse L cells and might block the MHV infection by non-specifically binding to surface proteins. On the other hand, among the three different time points of strictinin addition, the introduction of strictinin at two hours post-viral infection rendered the most pronounced effect on viral protein expression (Figure 6) and viral production (Figure 7), which also implicates the intracellular viral replication cycle that was effectively interfered with by strictinin. However, the precise mechanism of strictinin-mediated anti-MHV requires further investigation and evaluation using an animal model.

## 4. Materials and Methods

### 4.1. Cells and Virus

Both mouse L cell line and mouse hepatitis virus (MHV-A59) were kindly provided by Professor Hung-Yi Wu at the Graduate Institution of Veterinary Pathobiology, National Chung-Hsing University, Taiwan. Mouse L cells were grown in Dulbecco’s Modified Eagle’s Medium (DMEM, Gibco, Invitrogen, Carlsbad, CA, USA) supplemented with 10% fetal bovine serum (FBS, Invitrogen, Waltham, MA, USA) and antibiotics (penicillin 100 U/mL and streptomycin 10 μg/mL) at 37 °C with 5% CO_2_. The cells were washed with phosphate-buffered saline (PBS) prior to infection with viruses diluted in an infection medium (DMEM without FBS) supplemented with antibiotics.

### 4.2. Test Compounds

Theacrine (99% purity) was purchased from Bolise Co., Ltd. (Shanghai, China). Strictinin was purified as described previously [27]. Both of them were dissolved in double distilled water and stored at −80 °C. Remdesivir was purchased from Cayman Chemical (Ann Arbor, MI, USA), dissolved in DMSO, and stored at −80 °C.

### 4.3. Cytotoxicity Test

The PrestoBlue assay (Thermo Scientific, Waltham, MA, USA) was used to evaluate cytotoxicity [18]. Briefly, mouse L cells were seeded in 96-well plates (Corning Incorporated, New York, NY, USA) at a density of 9 × 10^6^ cells/well for 24 h. After the culturing medium was removed, the cells were washed with PBS twice prior to treatment with theacrine or strictinin of different concentrations (4, 20, and 100 μM) for 24 h. After removing the medium, the cells were treated with 10% PrestoBlue reagent for 1 h. The absorbance wavelengths at 570 nm and 600 nm were measured using a Tecan endless 200 PRO spectrophotometer (Tecan, Männedorf, Switzerland). The cell survival rate of the control treatment was set at 100%. The experiments were conducted in three independent repeats, and all data were presented as the mean ± S.D.

### 4.4. Viral Infection of Mouse L Cells with Different Treatments of Test Compounds

The effect of test compounds was evaluated by a time-of-drug addition assay following the strategy described in one previous report [46]. In brief, for a full-time treatment, a test compound was added to the DMEM for 8 h prior to MHV infection of mouse L cells, adjusted to 100 plaque-forming units (PFU) per well. For a pre-treatment, a test compound was incubated with mouse L cells for 8 h and then removed from the cells prior to viral inoculation. For a co-treatment, a test compound was mixed with MHV and then inoculated into mouse L cells for 2 h. After inoculation, the medium was removed from the cells. For a post-treatment, MHV was first inoculated into mouse L cells for 2 h, and a test compound was added to infected cells at 2 h post-infection (hpi) and remained through the duration of the infection. The experiments were conducted in three independent repeats, and all data were presented as the mean ± S.D.

### 4.5. Plaque Reduction Assay

Mouse L cells were seeded in 6-well plates (9 × 10^5^ cells/well) or 12-well plates (4 × 10^5^ cells/well). For a full-time treatment, cells were treated with theacrine or strictinin of 4, 20, or 100 μM 8 h prior to the time of infection, upon viral attachment, and during the course of infection. However, for the time-of-drug addition test [46], cells were treated with 20 μM of remdesivir or 100 μM of strictinin at three different time points, as described in Section 4.4.

In brief, for full-time treatment, cells were cultured in medium containing theacrine or strictinin for 8 h. After the medium was removed, the cells were washed twice with phosphate-buffered saline (PBS) prior to viral infection (100 PFU/well). Virus was diluted in medium containing the tested drug, and after incubation with cells at 37 °C for 2 h, the suspension containing unbound virus was discarded, and the cells were cultured for 2 days with DMEM (3 mL/well without FBS) containing 2% agarose (Lonza, Rockland, ME, USA) as well as the compound analyzed at 37 °C and 5% CO_2_. Infected cells were fixed with 100% methanol for 2 h and then stained with 1% crystal violet (Sigma-Aldrich, St. Louis, MO, USA) for counting viral plaques. The experiments were conducted in three independent repeats, and all data were presented as the mean ± S.D.

### 4.6. Western Blot Analysis

Proteins were denatured in boiling water for 10 min and separated by sodium dodecyl sulphate-polyacrylamide gel electrophoresis (SDS-PAGE; 5% stacking gel and 12% running gel) with a MINI-PROTEAN III apparatus (Bio-Rad; Hercules, CA, USA) [47]. Proteins resolved in SDS-PAGE were electrophoretically transferred to PVDF membranes (GE Healthcare Life Sciences, München, Germany). The membranes were incubated for 1 h at room temperature in PBS containing 0.1% Tween-20 and 5% dried milk. After removing the milk, the membrane was incubated with the primary antibody, i.e., mouse anti-nucleocapsid protein (1:4000 dilution) or anti-β-actin (1:1500 dilution) (Santa Cruz Biotechnology, Inc., Santa Cruz, CA, USA) in 10-fold diluted PRO Blocking Buffer (Energenesis Biomedical Inc., Taipei, Taiwan) at 4 °C overnight. The following day, the membranes were incubated with the secondary antibody (1:5000 dilution of peroxidase-conjugated AffiniPure goat anti-mouse IgG; Jackson ImmunoResearch Laboratories, West Grove, PA, USA) for 1 h. The signal was visualized with the Immobilon Western Chemiluminescent HRP Substrate (Merck Millipore, Madrid, Spain) after rinsing with PBS containing 0.1% Tween-20.

### 4.7. Statistical Analysis

All data were analyzed with GraphPad Prism 8 (Graph-Pad Software, San Diego, CA, USA), and presented as the mean ± standard (S.D.). The differences were analyzed by one-way analysis of variance (ANOVA), followed by post hoc analysis using the Tukey test to determine significant differences from the control group, where *p* < 0.05 was considered to be statistically significant.

## 5. Conclusions

Strictinin significantly inhibited MHV infection, particularly when it was administered upon viral attachment or after viral entry. These results suggest strictinin might be a suitable ingredient for the protection against and treatment of coronavirus infection. Given the above, the underlying mechanism of the pronounced inhibitory effect observed in the post-treatment of strictinin should be further elucidated by monitoring viral replication steps as well as the cellular signaling pathways regulated in the course of virus infection [46]. On the other hand, as strictinin could interfere with infection of MHV and influenza viruses, whether the broad antiviral activities of strictinin resulted from its non-specific binding to the surface proteins of various virus particles should be further validated. Moreover, as per previous reports, since infection with MHV could represent the disease observed in humans infected with HCoV-SARS and was regarded as a clinically relevant model of SARS [48,49], it merits investigating the effect of strictinin on MHV infection in mouse models. The information will be valuable for developing tea ingredients as an alternative option for the treatment of human coronavirus diseases.

## Figures and Tables

**Figure 1 molecules-28-01080-f001:**
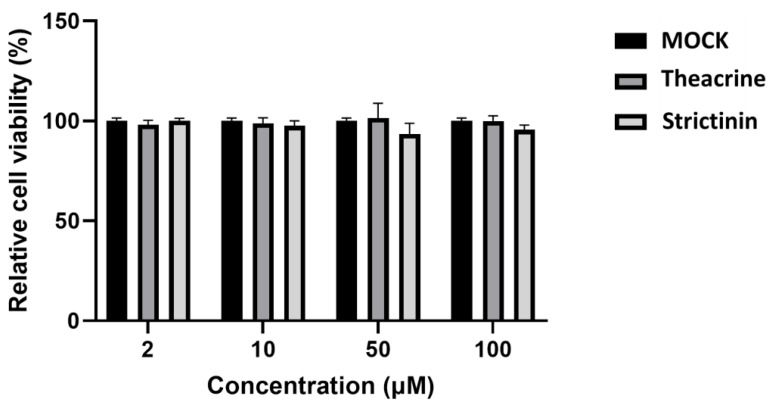
Cytotoxicity of theacrine and strictinin to mouse L cells. Mouse L cells were treated with theacrine or strictinin at concentrations of 2, 10, 50, and 100 μM for 24 h. MOCK was the control group. The cell viability was determined by PrestoBlue™ reagent. Data were displayed as mean ± standard deviation (S.D.) with each treatment in triplicate (*n* = 3).

**Figure 2 molecules-28-01080-f002:**
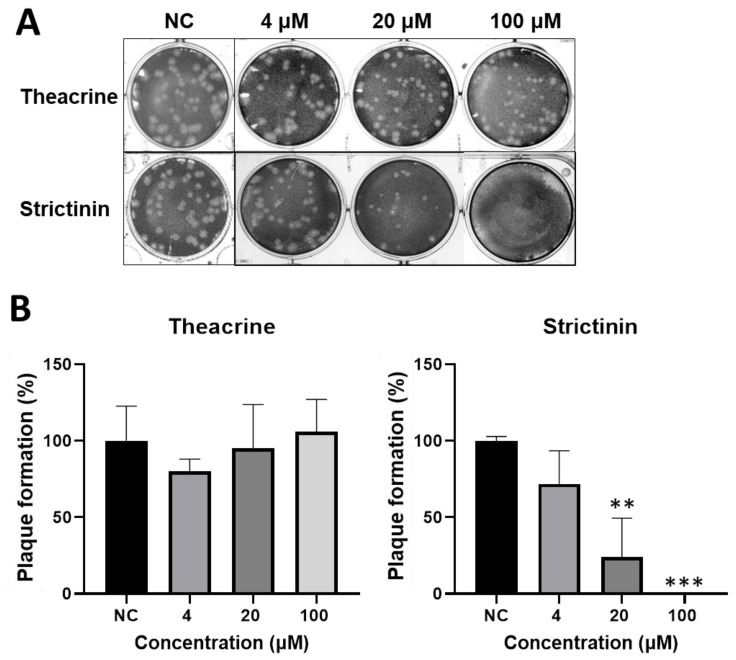
Evaluation of the effects of theacrine and strictinin on MHV infection of mouse L cells by plaque reduction assay. (**A**) MHV infection of mouse L cells was set at 100 plaque-forming units (PFU) per well. The infection was accompanied by theacrine or strictinin at a concentration of 4, 20, or 100 μM. NC was a negative control group with only MHV. (**B**) The relative numbers of virus plaques treated with theacrine and strictinin were calculated by using the plaque number in the NC group as 100%. All data were presented as mean ± S.D. (*n* = 3). ** *p* < 0.01 and *** *p* < 0.001, indicating significant difference from NC, were analyzed using one-way ANOVA, followed by Tukey.

**Figure 3 molecules-28-01080-f003:**
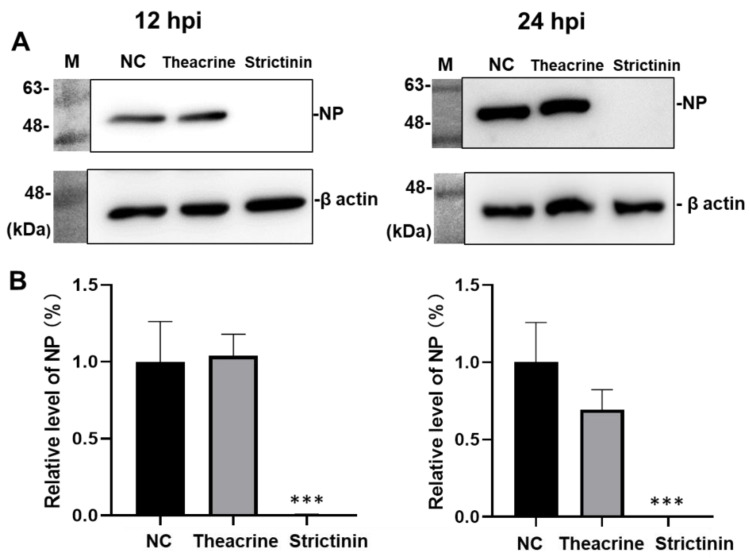
Effects of theacrine and strictinin on the level of nucleocapsid protein (NP) in mouse L cells after MHV infection. (**A**) MHV infection of mouse L cells was set at 0.01 multiplicity of infection (MOI) per well. The infection was accompanied by theacrine or strictinin of 100 μM. At 12 and 24 h post-infection (hpi), cell lysate was harvested, and the NP level was detected by Western blot analysis. NC was a negative control group with only MHV, and M was representative of marker proteins. (**B**) The relative expression level of NP in the NC group was set at 100%. All data were presented as mean ± S.D. (*n* = 3). *** *p* < 0.001, indicating significant difference from NC, was analyzed using one-way ANOVA, followed by Tukey.

**Figure 4 molecules-28-01080-f004:**
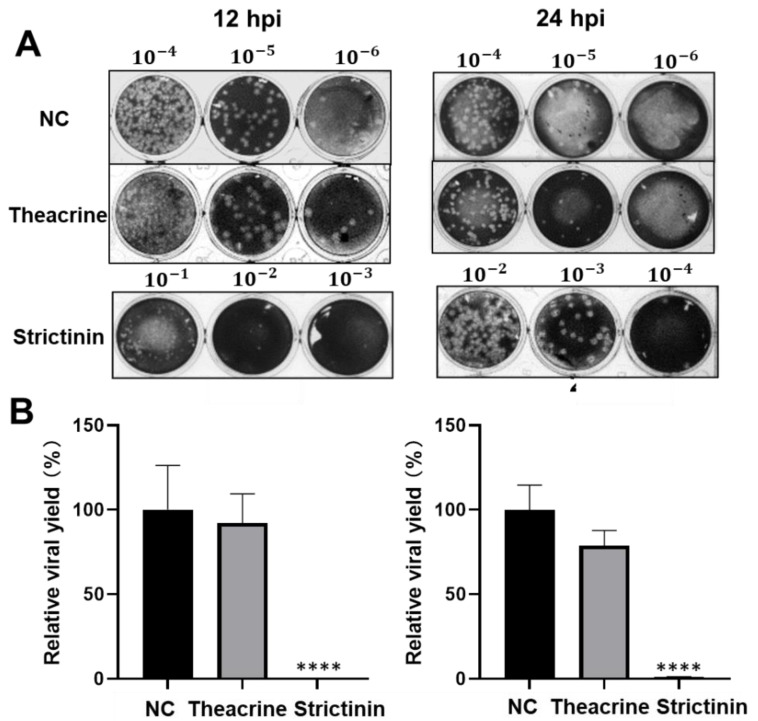
Effects of theacrine and strictinin on MHV progeny production. (**A**) MHV infection of mouse L cells was set at 0.01 MOI/well. The infection was accompanied by theacrine or strictinin of 100 μM. At 12 and 24 hpi, progeny production was estimated by the plaque assay of serial dilutions (indicated on top of the plates) of supernatants collected from infected cells. NC was a negative control group with only MHV. (**B**) The relative viral yield was calculated by using the plaque number in the NC group as 100%. All data were presented as mean ± S.D. (*n* = 3). **** *p* < 0.0001, indicating significant difference from NC, was analyzed using one-way ANOVA, followed by Tukey.

**Figure 5 molecules-28-01080-f005:**
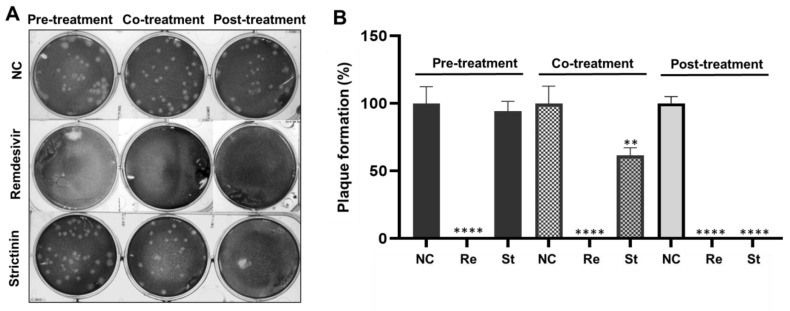
Evaluation of the effects of remdesivir and strictinin on MHV infection by plaque reduction assay. (**A**) MHV infection of mouse L cells was set at 100 PFU/well. A time-of-drug addition test was used in the plaque reduction experiment. Mouse L cells were pre-, co-, or post-treated with remdesivir (Re) of 20 μM or strictinin (St) of 100 μM. NC was a negative control group with only MHV. (**B**) The relative numbers of virus plaques treated with remdesivir and strictinin were calculated by using the plaque number in the NC group as 100%. All data were presented as mean ± S.D. (*n* = 3). ** *p* < 0.01 and **** *p* < 0.0001, indicating significant difference from NC, were analyzed using one-way ANOVA, followed by Tukey.

**Figure 6 molecules-28-01080-f006:**
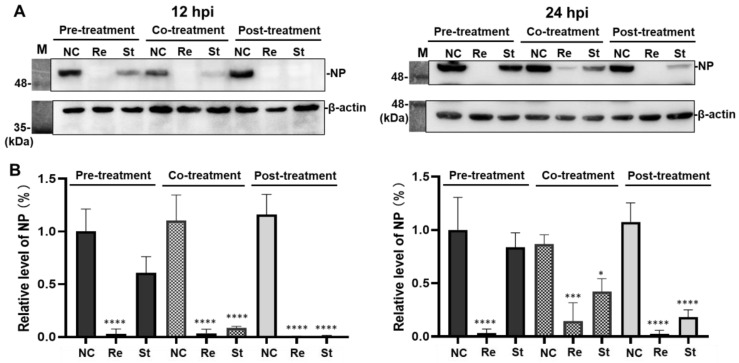
Effects of remdesivir and strictinin on the level of nucleocapsid protein (NP) in mouse L cells after MHV infection. (**A**) MHV infection of mouse L cells was set at 0.01 MOI/well. Mouse L cells were pre-, co-, or post-treated with remdesivir (Re) of 20 μM or strictinin (St) of 100 μM. At 12 and 24 hpi, cell lysate was harvested, and the NP level was detected by Western blot analysis. NC was a negative control group with only MHV, and M was representative of marker proteins. (**B**) The relative expression level of NP in the NC group was set at 100%. All data were presented as mean ± S.D. (*n* = 3). * *p* < 0.05, *** *p* < 0.001, and **** *p* < 0.0001, indicating significant difference from NC, were analyzed using one-way ANOVA, followed by Tukey.

**Figure 7 molecules-28-01080-f007:**
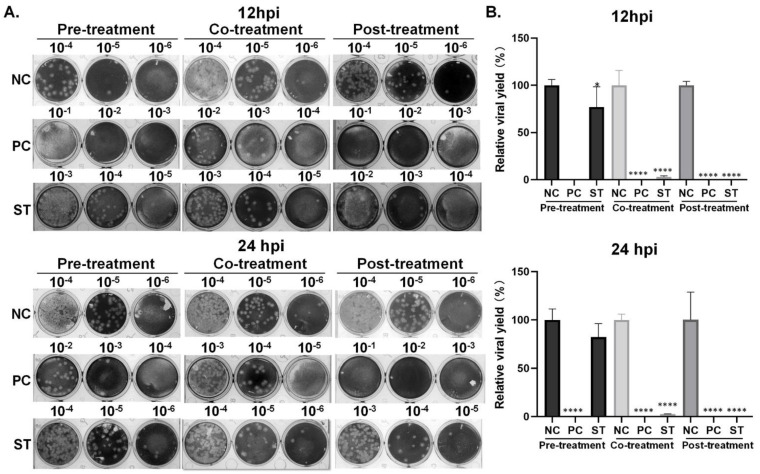
Effects of theacrine and remdesivir on MHV progeny production. (**A**) MHV infection of mouse L cells was 0.01 MOI/well. Mouse L cells were pre−, co−, or post−treated with 20 μM of remdesivir, as positive control (PC) or strictinin (ST) of 100 μM. At 12 and 24 hpi, cell medium was harvested, and progeny production was estimated by the plaque assay of serial dilutions (indicated on top of the plates) of supernatants collected from infected cells. NC was a negative control group with only MHV. (**B**) Relative viral yield was calculated by using the plaque number in the NC group as 100%. All data were presented as mean ± S.D. (*n* = 3). **** *p* < 0.0001, indicating significant difference from NC, was analyzed using one-way ANOVA, followed by Tukey.

## Data Availability

Not applicable.

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
