# Peer review of "Strictinin, a Major Ingredient in Yunnan Kucha Tea Possessing Inhibitory Activity on the Infection of Mouse Hepatitis Virus to Mouse L Cells"

_molecules, 2023, doi:10.3390/molecules28031080_

Round 1

Reviewer 1 Report

The manuscript of Erh-Chuang Tu, Wei-Li Hsu, and Jason T. C. Tzen "Strictinin, a Major Ingredient in Yunnan Kucha Tea Possessing Inhibitory Activity on the Infection of Mouse Hepatitis Virus to Mouse L Cells" describes the antiviral activity of strictinin on MHV. The manuscript is clearly written, and the experiments are adequate for the goals. However, I have some concerns regarding the introduction and the discussion. 

Major comments:

The introduction should give more information about strictinin and MHV, focusing on the latter's importance as model organisms. On the other side, the discussion is too generic and must be improved by discussing the results of each experiment, or most of them, in comparison with the literature. For instance, the authors performed a time-of-drug addition assay but did not discuss the data or the possible mechanism of action. Minor comments:

In material and methods, each described method should have an appropriate reference.

Reviewer 2 Report

To,

The Editor,

Molecules, MDPI,

Manuscript ID: molecules-2122634

Subject: Submission of comments of the manuscript in “Molecules"

Dear Editor Molecules, MDPI,

Thank you very much for the invitation to consider a potential reviewer for the manuscript (ID: molecules-2122634). My comments responses are furnished below as per each reviewer’s comments. 

In the reviewed manuscript, the authors evaluate the effects of these two compounds on the infection of mouse hepatitis virus (MHV), an important -coronavirus pathogen for mice, to Mouse L cells. Treatment of strictinin (100 μM), but not theacrine, completely eliminated MHV infection, including plaque formation, nucleocapsid protein expression and progeny production of MHV. Subsequently, remdesivir, a potential drug for the treatment of SARS-CoV-2 was used as a positive control to further evaluate the possible mechanism of antiviral activity mediated by strictinin via time-of-drug addition protocol, including pre-, co- or post-treatment, against MHV infection. The results showed that all the three treatments of remdesivir (20 μM) completely eliminated MHV infection. In contrast, no significant effect on MHV infection was observed for the pre-treatment of strictinin (100 μM) while significant inhibition or complete elimination of MHV infection was observed for the co-treatment or post-treatment of strictinin. Evidently, strictinin significantly inhibited MHV infection and Pu'er tea rich in strictinin might be suitable supplement for the protection against coronavirus infection.. In general, the manuscript represents a very big piece of information in a logical presentation. Therefore, it might be conditionally accepted subject to minor revision. Authors have to improve their manuscripts with many non-clear meanings, inaccuracies and inconsistencies, and the authors need to address the following issues before it can be accepted for publication.

  1. I have read the entire manuscript and my initial comment is that manuscript is poorly written. I have significant concerns about the grammar and vocabulary of the manuscript; therefore, I recommend the authors to used an English proofreading service.
  2. The structure of the abstract should be improved, as well as the lack of several aspects that should be included in this section. Most of the abstracts contain confusing and uninformative sentences. Please give more precise objectives here (such as in the Abstract). The abstract should highlight the most important results of the parameters and characteristics assayed.
  3. Introduction grammatical issues appear to be most prevalent in the introduction, making for very confusing reading. Further, the introduction is short but has no clear thread.
  4. Figures 7A have quite low resolution and difficult to make out. Higher-resolution versions will be needed for publication, Further, figure texts are not readable.
  5. In Material and Methods:- indicate how many replicates assayed in each analysis/parameter. The number of samples or biological and technical replicates should be mentioned for each parameter in the methods.
  6. Results must be explained clearly and in detail.
  7. The discussion should be interpreted with the results as well as discussed in relation to the present literature.  
  8. Authors must add the conclusion section.

Reviewer 3 Report

This manuscript describes anti mouse hepatitis virus activity of strictinin, a major ellagitanin in Yunnan Kucha tea.
The antiviral activity of strictinin against Mouse Hepatitis Virus (MHV) has not been reported. On the other hand, since anti-influenza virus activity is already known, it is reasonable to speculate that strictinin has anti-MHV activity. Therefore, simply reporting the effects on MCV infection and proliferation in mouse cells is not scientifically novel. For example, it is necessary to conduct experiments to clarify the mechanism of anti-MCV action to some extent, or to clarify the in vivo action in mice, to analyze in detail the findings on the anti-MCV action of strictinin, and to enhance the scientific novelty of the paper.
It has already been reported that corilagin, a type of ellagitannin with a very similar structure to strictinin, shows anti-MHV activity (Ref. Fitoterapia 2014, 99, 117-123). In this regard, it is necessary to consider the relationship between the chemical structure and the anti-MHV action.

Round 2

Reviewer 3 Report

Since there have been no revisions to any of the points I raised in this revised manuscript, I will not change my judgment of this paper.
Even if MHV is a different type of virus from influenza viruses, it is quite plausible that strictinin has anti-MHV activity, since corilagin, a type of ellagitannin with a very similar structure to strictinin, has already been reported to exhibit anti-MHV activity. Therefore, I judge that the content of this paper is not novel enough to merit publication in Molecules, although I do not say that there is no scientific novelty at all.
The degree of novelty required to merit publication in Molecules is not mine alone to judge, and I leave it to the editor to make that judgment.
